# Augmented Reality (AR) and Cyber-Security for Smart Cities—A Systematic Literature Review

**DOI:** 10.3390/s22072792

**Published:** 2022-04-06

**Authors:** Nouf M. Alzahrani, Faisal Abdulaziz Alfouzan

**Affiliations:** 1Information Technology Department, Collage of Computer Science and Information Technology, Al Baha University, Al Bahah 65731, Saudi Arabia; noufalzahrani@bu.edu.sa; 2Department of Forensic Sciences, College of Criminal Justice, Naif Arab University for Security Sciences (NAUSS), Riyadh 14812, Saudi Arabia

**Keywords:** augmented reality, cybersecurity, smart city, systematic literature review

## Abstract

Augmented Reality (AR) and cyber-security technologies have existed for several decades, but their growth and progress in recent years have increased exponentially. The areas of application for these technologies are clearly heterogeneous, most especially in purchase and sales, production, tourism, education, as well as social interaction (games, entertainment, communication). Essentially, these technologies are recognized worldwide as some of the pillars of the new industrial revolution envisaged by the industry 4.0 international program, and are some of the leading technologies of the 21st century. The ability to provide users with required information about processes or procedures directly into the virtual environment is archetypally the fundamental factor in considering AR as an effective tool for different fields. However, the advancement in ICT has also brought about a variety of cybersecurity challenges, with a depth of evidence anticipating policy, architectural, design, and technical solutions in this very domain. The specific applications of AR and cybersecurity technologies have been described in detail in a variety of papers, which demonstrate their potential in diverse fields. In the context of smart cities, however, there is a dearth of sources describing their varied uses. Notably, a scholarly paper that consolidates research on AR and cybersecurity application in this context is markedly lacking. Therefore, this systematic review was designed to identify, describe, and synthesize research findings on the application of AR and cybersecurity for smart cities. The review study involves filtering information of their application in this setting from three key databases to answer the predefined research question. The keynote part of this paper provides an in-depth review of some of the most recent AR and cybersecurity applications for smart cities, emphasizing potential benefits, limitations, as well as open issues which could represent new challenges for the future. The main finding that we found is that there are five main categories of these applications for smart cities, which can be classified according to the main articles, such as tourism, monitoring, system management, education, and mobility. Compared with the general literature on smart cities, tourism, monitoring, and maintenance AR applications appear to attract more scholarly attention.

## 1. Introduction

Rapid technological advancement has taken place in the last decade, propelled by developments and advances in information and communication technologies (ICT). This has revolutionized the way people communicate, live, work, and travel, among other things. As a result, smart cities have emerged, evolving towards intelligent, dynamic infrastructures that serve civilizations while accomplishing the criteria of sustainability and energy efficiency [1]. According to [2,3], the meaning of smart cities implies the integration of existing substructures with novel ICTs to produce an all-inclusive system of efficient urban services. Correspondingly, Ref. [4] note that a smart city is a conurbation that connects physical infrastructure, business infrastructure, social infrastructure, and information technology (IT) infrastructure to strengthen the city’s collective intelligence. Although there exists no general consensus on the definition of a smart city, there is a unanimity within literature that one of the main goals of smart cities is to improve the quality and efficiency of city services, while at the same time making better use of public resources and reducing operational costs [5,6]

One of the new technologies that are widely deployed and exploited within smart cities is Augmented Reality (AR), which, as noted by [7], can enhance human-machine interaction. Immersive technologies, such as AR and virtual reality (VR), which either superimpose digital content into a physical world or immerse users into an altogether different, interactive, and digital environment, are shown to have exceptional and convincing uses in smart cities. As discussed by scholars such as [8,9,10], the availability of high-speed and consistent network connectivity has enabled AR technology to mature enough to impact cities in becoming smart, digital, and connected in various ways, including disaster response, enabling medical services, and navigation management. Related technologies such as VR have also enabled services and programs such as police training, education, and urban planning. In particular, Ref. [11] mention that AR can be described as a coincidental combination between virtual objects and the real world, which enables real-time interaction as well as three-dimensional virtual registration.

With human perception regarding the environment increasingly changing with modern technology, AR becomes progressively prominent. Ref. [11] further note that AR adds virtual information to a real environment, ultimately impacting user cognition. This augmentation of the virtual intangible information into the tangible world impacts how people live and interact in smart cities. The application of AR into smart cities offers a unique immersion into the Internet of Things (IoT) applications. Recent research suggests that this can guide an interactive demonstration of how public services such as street control [12] video surveillance [13] solid waste collection [14], and parking management [15] can be accomplished and controlled from a single platform, making cities safer, cleaner, and more livable. As suggested previously herein, the key objective of smart cities is to connect everything together and to people. Consequently, AR technology enables smart city inhabitants to have an instant and immersive connection with everything around them.

The use of smart technologies such as AR, however, raises new issues and challenges. In smart cities, the vulnerable action of users and organizations can put the entire city at risk of cybercrimes. Ref. [2] notes that due to the reliance of various components of smart cities on ICT, cyber-security challenges, including malicious cyber-attacks and leakage of sensitive information, may affect a smart city’s behavior. When smart cities lose control of their technologically developed systems, it can negatively impact quality of life, people’s security and privacy, the city’s economy, technological axis, and even more, can put people at risk [16]. Given this, in order to respond to the increasing fervent acceptance of global smart city technologies such as AR, cyber-security programs must be developed in the same direction. Congruently, Ref. [17] argue that it is clear that cyber-threats for smart cities need to be taken extremely seriously. The researchers identify key areas for possible solutions, including the development of procedures and action plans for responding to cyber-attacks; the implementation of manual overrides and failsafes on all smart city systems; and the creation and use of security checks for encryption, authorization, authentication, as well as software updates while implementing new smart city systems.

In this work, an overview of AR applications in smart cities around the world is provided. The paper also investigates the topic of cyber-security for smart cities, demonstrating how specific aspects of smart cities give rise to cyber-security challenges in an augmented reality world. In other words, this article focuses on cyber-security risks that may affect data privacy and the outputs generated by the AR applications through the device, and how this could impact the implementation of AR technology in smart cities. The paper adopts a systematic literature review approach in order to achieve a comprehensive research synthesis on the basis of evaluation and analysis of literature under investigation in this study. In the end, it is believed that the findings of this study will have implications for academicians, practitioners, and societies at large. It also adds to the academic literature on the use of AR technology in the building of smart city infrastructure, where government officials and urban planners can use this technology to enhance cities’ environments, infrastructure, services, and the quality of life of urban dwellers.

The study consists of six parts: Introduction, Smart City Concept, Methods, Results, Discussion, Conclusion and Future Work. The second chapter describes the concept, elements, and general structure of a smart city. The third chapter presents the methodology adopted in conducting the systematic literature review, including the search strategy and inclusion and exclusion criteria. The fourth chapter presents the results of the review, while the fifth presents the discussion. In the final chapter, the conclusion and future research are stated.

## 2. Smart City Concept

As noted by researchers such as [7], smart cities describe cities that use ICT systems to disseminate information to citizens, enhance operational efficiency, and ultimately improve the quality of public services. In their view, some key aspects of smart cities include smart education, smart governance, smart environment, smart homes, smart mobility, smart energy, and smart health. In terms of energy, Ref. [18] agree that smart cities can have the ability to control and monitor the amount of energy consumed and distributed using ICT technologies, which can enable cities to improve reliability and provide greater power quality and profit growth. When it comes to health, Ref. [19] note that the monitoring of people’s health through IT technologies such as sensor devices help cities provide real-time information on patient health indicators (breathing, temperature, heartbeat), enabling faster and better decisions. Ref. [18] further note that environmental parameters such as air quality, humidity, and temperature are vital for smart cities. As such, ICT technologies such as AR can aid in the management of such parameters, with applications already successful in areas such as water and air quality and garbage management. In terms of traffic and mobility, Ref. [19] argue that efficient exploitation of IoT technologies can help in solving the problem of traffic congestion, as well as issues in existing transport infrastructure.

The infrastructure of a smart city can create a unique collaborative system where citizens, educational institutions, industries, prosumers, and researchers can develop innovative products, services, and solutions. Contrary to traditional double-sided marketplaces, Ref. [20] argue that the ecosystem that results from a smart city allows a multitude of actors to be engaged in private and public consumption, production, professional activities, entertainment, research, and education. As suggested herein and noted by [17], a smart city can be divided into six fundamental aspects: smart people, smart living, smart mobility, smart government, smart environment, and smart economy (see Table 1), with each element having its key indicators and benefits.

Presently, there are many existing urban projects that have been transformed to meet the aforementioned smart city criteria. Some projects have been developed from scratch, while others consist of the transition and modernization of existing cities into smart cities. As noted by [1], some of the worldwide smart cities that have been developed from scratch include Skolkovo in Russia, PlaIT Valley in Portugal, Lavasa in India, Masdar City in Abu Dhabi, Meixi Lake in China, and Songdo in IBD. Some of the elements and features of such cities include the development of large tech-driven business centers, innovative education, cultural and medical services to citizens, sustainable urban infrastructure, green buildings, power supply through renewable energy, and collective broadband connectivity. Smart cities that have been developed from existing urban centers include London, Santander, and Portland. According to [1], some actions that have been undertaken by urban developers to make such cities smart include the installation of intelligent systems for waste collection and management, implementation of payment systems through smartphones, control and monitoring of public security through video monitoring technologies, implementation of smart devices for tracking and monitoring people’s health, improving tourism through interactive mobile applications and immersive technologies such as AR and VR, and management and control of traffic through ICT technologies.

## 3. Methods

In order to achieve this study’s objectives, a Systematic Literature Review (SLR) methodology was followed. As pointed out by [21], the SLR approach aims at searching, appraising, synthesizing, and analyzing all studies relevant to a specific field of research. Given the nature of this study, the SLR research approach was adopted to aid in planning, searching, screening, extracting data, and synthesizing and reporting findings.

### 3.1. Search Strategy

One of the fundamental goals of this study was to be as inclusive as practically possible. However, the idea was also to obtain scholarly papers within the last decade to ensure that materials gathered were the latest, and coincided with the digital age and era of smart cities. As such, all papers published in journals and conferences between 2010 and 2021, which included the phrases ‘Augmented Reality for Smart Cities’ and ‘Cyber-security for Smart Cities’, and related user studies were considered. In the planning phase, some of the most utilized online scientific databases were used to search for peer-reviewed literature. Three relevant literature databases were selected, including Emerald Insight (EI), Science Direct (SD), and IEEE Xplore (IX). As research sources, these multidisciplinary databases were selected and recognized for their indexing and coverage. They were consulted, and subsequently, the results obtained were cross-checked. Furthermore, the databases used in these academic studies have met the protocol requirements and used protocol-specific parameters.

### 3.2. Inclusion and Exclusion Criteria

The results from the search process were screened against pre-set inclusion and exclusion criteria, as mentioned in Table 2. In terms of inclusion criteria, only studies published between 2010 and 2021 were considered to ensure that interventions and applications related to AR and cyber-security for smart cities were relevant and up-to-date. Secondly, studies were only considered if they reported the application of AR and/or cyber-security for smart cities. As such, studies that reported the advantage, limitations, uses, challenges, effectiveness, and scope of AR and cyber-security in smart cities were considered for inclusion. Only studies that were published in the English language were considered for this review’s inclusion for practical reasons.

Among the exclusion criteria, this study excluded studies that only reported on AR and cyber-security initiatives in other contexts. Studies that did not mention AR or cyber-security in smart cities were also excluded. The same applied to studies that claim to report AR but referred to VR or mixed reality (MR) instead. Studies not considered as peer-reviewed journals, book chapters, or conference papers in the context of AR and cyber-security in smart cities were excluded.

### 3.3. Search Results

After performing the search based on the identified keywords, as shows in Figure 1, 421 documents were initially identified from the three selected scientific databases. The initial search was done on 6 September 2021 and the last was done on 14 September the same year. An initial filter was conducted on the identified 421 studies among scientific articles, book chapters, and conference articles. This included evaluating the inclusion and exclusion criteria, considering the titles and abstracts of studies, and cross-checking the results of the three databases to remove duplicates. After this initial filter, 176 studies remained.

After further screening titles, abstracts, introductions, and conclusions, a further 105 studies were removed. Finally, following full-text reading and examination of each of 71 articles, a total of 31 studies met the criteria proposed and definition for inclusion in this review.

### 3.4. Data Extraction

This phase was a two-part process in which each individual article was analyzed to determine final inclusion into the SLR. To minimize bias, this process was conducted by two individuals, each screening articles independently. The initial part involved skimming through the title and abstracts, as well as introductions and conclusions, to determine whether the obtained articles were relevant in addressing this study’s research objective. The second stage involved skimming through and subsequently reading through full texts to obtain relevant data for SLR. In this stage, five QA criteria were developed to assess each study’s quality.

Q1: what was the application area (education, construction, tourism, design, or healthcare)?Q2: what was the type of data collected?Q3: what was the experimental design (methodology)?Q4: where was the research conducted (based)?Q5: what type of study was conducted (case study, field, experiment, formal, or pilot)?

In order to systematically and accurately record data based on these QAs, an excel spreadsheet was developed. During the review of the paper and data extraction, the researcher also flagged certain publications (especially in references) for additional discussion and cross-referencing. The data extraction form only contained information relevant to this study for analysis and descriptive purposes (in the discussion chapter) later in the SLR.

### 3.5. Synthesis

Since this SLR spans across a number of academic disciplines and fields, a thematic synthesis approach was adopted as the modulus of analysis. As discussed by [22], this approach in data synthesis is particularly suitable for analyzing and synthesizing multidisciplinary datasets. To address the key aims of this study, common themes across the included studies were identified and analyzed in detail. The identified concepts of AR and cyber-security applications were seen as the starting point from which to introduce a common language to compare and contrast identified perspectives and findings in included studies. As the common concepts in this study, AR and cyber-security applications and interventions in smart cities were used to provide a common denominator for developing new themes from the included studies.

## 4. Results

The search process produced 31 studies that were included in the SLR. The ensuing sections in this chapter provide a summary of results according to the pre-stated data extraction criteria intended to fulfill this study’s research objectives. One of the findings is that most of the studies related to AR and cyber-security for smart cities contexts are mostly published in Science Direct (16) and IEEE Xplore (12). The third database for this investigation, Emerald Insight, only produced three results; two relating to AR and one to cybersecurity, under the researched contexts. The majority of included studies were published in 2019 (14). Of the rest, three were published in 2014, three in 2016, five in 2017, seven in 2018, and nine in 2020. Based on this SLR, the number of publications increased from 2014 to 2020. This drastic increase may have been caused by technological advances and the development of smart cities. Between 2018 and 2020, there is particularly a consistent number of publications related to AR and cybersecurity for smart cities. As highlighted in the subsequent sub-sections, AR and cybersecurity papers for smart cities are divided into various applications and relating aspects.

### 4.1. AR for Smart Cities

Results under this segment indicate that the most discussed potential AR applications for smart cities are in various categories including, tourism, information dissemination, mobility, risk management, construction, education, energy management, and traffic monitoring. These application categories can be further grouped into five distinct classifications, including healthcare, robotics, public sector, tourism, (system) monitoring, (system) management, education, and mobility. Through this SLR, these categories have been critically compared based on aim, methodology, and chronological perspective. The results indicate that a number of AR application studies for smart cities focus on tourism and management/monitoring aspects comparing to other aspects.

#### 4.1.1. Tourism

There is considerable consensus that AR allows for the simultaneous perception of the real environment and a virtual audio overlay. This is particularly vital in mobile applications, where users are continuously aware of their surroundings, such as in the case of urban smart and urban tourism, where travelers explore foreign sights and cities. Recent studies report that AR continues to become increasingly popular within the travel industry, especially because it enables attraction sites, hotels, and businesses operating in this industry to enhance physical environments while encouraging both local and international tourists to visit. As seen subsequently in the Discussion Chapter, previous studies in this area have shown that AR can be used in a variety of ways, including augmented reality gasification, beacon technology and push notifications, augmented reality destinations, and interactive hotel and attraction elements. However, AR application in tourism for smart cities is not well explored, and as such, publications relating to the same are limited. From database search, this study only obtained three studies: [23,24,25].

Ref. [23], in his paper on smart tourism, discusses issues, challenges, and opportunities presented to the industry by smart technologies such as AR. In this paper, the scholar clarifies that the smart concept signifies the integration of organizational networks and smart features such as AI, IoT, big data VR, and AR to enable automation, facilitate daily activities for users, and enrich the ecosystem and way of life. These latest technologies have given rise to the concepts of ‘smart destination,’ ‘smart city,’ and ‘smart planet’ that have become increasingly popular in recent times. Thanks to the visualization feature, the researcher argues that AR application, in particular, has enabled smart cities to increase the quality of tourist experience while also enhancing interaction with the physical world. During on-site travel experience, it is noted in this paper that AR can provide information about the destination, including image recognition platforms, multiple viewpoints of the destination, and landscape information that can be viewed in adaptive screens. Given the power of this technology, Ref. [23] notes that it can be applied in various areas, including accommodation enterprises, F&B businesses, and museums, among others.

Correspondingly, Ref. [25] believe that advances in AR are expected to further push the boundaries of what data can be collected and how they can be utilized to improve touristic experiences for smart cities. They argue that, in connection with the physical infrastructure (e.g., smart tourism and smart city), the focus is on blurring the lines between the physical and the digital, and fostering digital integration. In the context of tourism, smart technologies such as AR are essential in altering consumer experience and are fostering creative tourism business models. In connection with other technologies such as Social Networking Services (SNSs), VR, beacon technology, geo-tag services, location-based technologies, mobile apps, big data, and cloud computing, AR is enabling businesses in smart cities new ways of advertising, novel collaborative ventures, better tourist services, and better ways of managing tourist flows to innovate beyond traditional industry boundaries. Ref. [25] further note that smart tourism powered by these technologies allows travelers to better interact and communicate with and in cities, and to establish closer relationships, not only with residents but also with city attraction highlights, local government, and local businesses.

In the third paper for AR application in tourism for smart cities in this study, Ref. [24] explore an AR-based prototype in the pilot region of Gökova Mugla, Turkey. In their study, a mobile application prototype was developed using AR aimed at introducing sightseeing places, hotels, restaurants, touristic destinations, and other important centers for both domestic and international tourists. Key information such as price, social media data, comments, ratings, and intensity about these areas was simultaneously provided on the mobile application, while location data and image processing techniques were also used for implementing AR technology. The researchers suggest that a successful implementation of such an application can be important for smart tourism in a number of ways. First, social media integration can increase an attraction’s recognition and popularity. Second, AR-based applications can remove the need to carry maps and brochures, allowing tourists to obtain real-time information conveniently. Third, such an application can provide current and simultaneous instant price, intensity, position, and evaluation information, while also allowing extra virtual images with actual images simultaneously.

#### 4.1.2. System Monitoring

Performing monitoring and maintenance tasks in complex environments has long been shown to be challenging and difficult due to complexity, and possibly due to the use of multifaceted processes, heavy machinery, human factors, uneasy access, and underground facilities, among others. The underlying technology is often shown to be inefficient because of simultaneous supervision of people working together under extreme settings, missing multi-input interfaces, and significant delays in communication and data transmission. Recently, however, AR as a tool for information visualization is said to provide solutions to these issues, where real-time reports are essential for monitoring systems and aiding in decision making. This SLR identified two applications where this technology can be applied for smart cities. The first is the study of [26] that analyses the application of AR service for efficient information dissemination based on a deep learning algorithm for the smart city of Jeddah in Saudi Arabia. Their proposed framework uses the system architecture of the iMARS system that consists of user alerts, deep learning, databases, and municipality services. These entire components are correlated with each other to enable the application to provide real-time and prioritized information to users, while also aiding governmental departments and other organizations to prioritize services based on user needs.

In the second paper under this category, Ref. [8] explore the application of AR for traffic monitoring in smart cities. By integrating a number of modern technologies, including AR, cloud computing, machine learning, Internet of Vehicles, and IoT, the researchers intended to develop a smart traffic control system that can produce traffic updates as well as road status based on the strength of vehicles in smart cities. They used two types of sensors: roadside sensors to provide information about the condition of the road, and vehicle sensors to keep track of the entire information of the vehicle. VEINS (Vehicle in Network Simulator), which is an amalgam of network simulator OMNET++ and road traffic simulator SUMO, were used for analyzing traffic density according to different simulations. Their proposed Internet of Vehicle with AR for smart traffic monitoring model indicates that AR-based systems can effectively be used for smart transportation and real-time traffic monitoring.

#### 4.1.3. System Management

This study identified four studies that examine how AR can be used in management systems in various areas, including rent management, emergency management, lighting system management, and energy management. In the first study under this set, Ref. [27] analyze how visualization by AR can be used for smart rent portals in smart cities. Their study proposes a recommender system that is managed and visualized through AR and Vuforia to provide a platform that allows users to hold out a preference-based cooperative filtering search on rental properties. This recommender system is shown to enable rental service seekers to refine their preferences based on shallow learning. This study is grounded on the premise that locating services or products online that meet every users’ preference is more and more troublesome, owing to the massive pool of decisions to think about before inbound at the required choices. With a management system powered by technologies such as AR, the researchers conclude that users in smart cities can now be able to visualize products and services (such as rental properties) to enable more informed decisions than traditional frameworks and algorithms.

Similarly, Ref. [28] illustrate how the same can be applied for emergency management in smart cities. In their review of relevant literature through the Web of Science Core Collection and snowballing, the scholars note that immersive technologies such as AR and VR can be applied to emergency management systems to allow for better response to emergency situations that may arise in cities. They provide an instance where AR can be used combined with databases to explore how to efficiently maintain fire safety equipment and cope with related fire hazards. They argue that AR can serve as an effective approach through which disaster scenes can be reproduced in a way that information can be extracted from the virtual setting for safety assessment, and research results can be applied in real settings. Given the potential this technology has, Ref. [28] further note that it can be used in emergency simulation systems to study potential dangers in specific situations such as potential coastal flooding, evacuation safety, and productivity in manufacturing plants, oxygen deficiency in the steel industry, as well as accidental perception in construction sites.

A wireless, AR-powered LED streetlight system with centralized and remote-control technology has also emerged as an innovative application for smart cities. In their study, Ref. [29] illustrate how such a lighting management system with remote control capability can provide numerous benefits for smart cities, including monitoring and real-time control capabilities, as well as reduced energy consumption and operational costs. In their study, they present two forms of applications powered by modern technologies such as AR and IoT to aid in lighting management for a learning institution. The first is a mobile application that can generate the safest walking paths on campus by integrating streetlights with various pedestrian-counting video sensors. The second is an emergency response aid application that integrates streetlights with on-campus 911 emergency buttons. Both applications were designed specifically with the goal of public safety improvement. By integrating visualization and intelligence into such systems, the scholars suggest that this can enable controlling of light brightness in real-time based on environmental dynamics to lower power consumption, while also managing the schedule of light systems. Maintenance of the streetlights can also be done efficiently by relying on a network system with sensors that provide usage statistics and operational states.

The search results also indicated that AR could be used in energy management for human-computer interaction in a smart city. Traditional energy management systems are often based on graphical and numerical values through monitor screens. Ref. [30] suggest that such systems are prone to errors and are often very difficult for consumers to interpret. As such, the researchers propose a novel, AR-enabled interface that can easily be accessed and interpreted by users in smart cities. Through a diorama that can visualize energy data intuitively, Ref. [30] illustrate how this new EMS system can enable users to check operation, renewable energy production, energy consumption, as well as environmental information of the zone where the systems are installed through Augmented Reality Interface (ARI). Such information can then be used for improving the indoor environment (maintenance) and reducing unnecessary energy consumption.

#### 4.1.4. Education and Instruction

The combination of educational content with AR technology to create a new type of automated applications and enhance the effectiveness and attractiveness of teaching, learning, and instruction has been explored widely in recent times. However, this exploration is yet to attract considerable attention under the smart city context. In fact, this study’s SLR only retrieved two studies from the selected databases that met the inclusion criteria. The first was that of [9] who attempted to explore an AR application for smart campus urbanization using a Mugla Sitki Kocman University (MSKU) campus prototype. Although their study falls under the education and instruction category in this study, their application was not designed to aid in learning or instruction. Rather, they suggested a platform that can offer more technical services to users in the university, improve campus technological infrastructure, increase the interaction between the campus and students, improve training services and presentation of technological learning environment for the learner, and enable easier identification of buildings and other locations in the campus.

In the second paper under this category, Ref. [31] examine how AR can be used to present work instructions for factory employees in order to lighten the workload of employees (particularly those in assembly, maintenance, and set up) and boost factory efficiency. The researchers developed a new smart solution for designing and presenting work instructions through AR, including virtual instructions on the screen (with or without special controllers, or with or without in-situ projections), video instructions, and traditional ‘paper’ instructions. These are designed as a software system for developing and working with virtual assembly instructions. Ref. [31] designed and tested this software capable of creating 3D animated assembly instructions. The results suggest that this software can provide easy and efficient ways for both technical and non-technical employees to receive instruction to improve their work efficiency and productivity.

#### 4.1.5. Mobility

Mobility, although examined limitedly, is another area of application of AR in smart cities. This study only produced one study that explores this application in the smart city concept. Ref. [32] investigated how AR and IoT can be used to improve the accessibility of people with motor disabilities in smart cities. The researchers designed a system that enabled wheelchair users to interact with items placed beyond their arm’s length with the help of Radio Frequency Identification (RFID) and AR. This application was an interactive AR that ran on different interfaces to enable users to digitally interact with physical items thanks to the updated inventory by an RFID system. The result of the study suggests that an AR-powered system can not only enable disabled people with wheelchairs to interact with the physical world, it can also enable them to visit and experience various site activities in an autonomous way.

### 4.2. Cyber Security for Smart Cities

The implementation of technologies such as AR and IoT in smart cities is normally hailed as the solution to numerous urban problems such as environmental protection, waste and energy management, and transportation. However, the implementation of these technologies is seen to be the source of numerous cybersecurity issues. Scholars agree that as smart cities become increasingly interconnected and the level of digital infrastructure becomes more complex, such cities will become more vulnerable to cyber-attacks. This SLR produces 17 publications that address the topic of cybersecurity for smart cities, with the majority of articles published in the Science Direct database. After reviewing the articles, the researcher classified the results under two main categories: challenges (issues) and opportunities (solutions).

#### 4.2.1. Challenges/Issues

One of the key papers under this category is from the Oxford Analytica Daily Brief that analyses industrial [33], business, social, economic, and geopolitical developments on a global and regional basis, providing various sectors with a timely and authoritative analysis of various issues and solutions. According to this brief, the cyber security of smart cities will be critical in harnessing the cost and efficiency gains brought about by increased connectivity. It is believed that systems that analyze and interpret consumer behavior in these cities will continue to present unique challenges of security and privacy, as they create attractive targets for malicious actors, intelligence agencies, and repressive regimes. This expert briefing further states that cybersecurity protections will be a point of differentiation among technology vendors; citizens will continue to bear ultimate responsibility for what data they share and how they are used, and innovation among technology vendors will be inhibited so long as they see a risk of liability when collaborating with cities. Similar challenges have also been reported in [34,35,36,37].

In a systematic literature review, Ref. [2] also identified a number of security threats and problems that may affect smart cities—first, eavesdropping on information sent from sensors and equipment, giving cyber-security attackers sufficient knowledge to undertake malicious acts. Second, distortion of messages sent to subsystems. As a result, incorrect messages are sent to various devices, and the normal operation of systems is disturbed. Third, attackers are also able to delay the messaging and communication between systems to have the effect of denial of service. Some of the messages exchanged in the transmission and distribution systems may be time-critical and must be transmitted within a short period of time. When this does not happen, major negative repercussions may follow.

Reference [38] provide a thorough insight into the smart city threat landscape for components related to acquisition and storage of smart city data emerging from components such as smart vehicles, unnamed aerial vehicles, building automation systems, and smart grids, along with enabling technologies such as cloud computing and IoT sensors. From the cloud, they identify a variety of cybersecurity issues, including system and application vulnerabilities, malware injection attacks, denial of service (DoS), malicious insider threats, and data leakage. From IoT sensors, a number of issues can also materialize, such as remote exploitation, sensor failure, data storage, and management problems, insecure communication, and confidentiality leaks. With smart grids, problems such as the attack on internet-connected devices, rogue/infected devices, eavesdropping, privacy, and protocol vulnerability can also occur.

Reference [38] also identify a variety of physical threats from such systems, such as introducing data glitches to gain unauthorized access to debug interfaces, side-channel attacks to leak information, or fault-injection into the ECU to defeat central locking systems. Other scholars including [39,40] also report similar cybersecurity challenges for smart cities, but further address other issues such as man-in-the-middle attack, phishing, and spoofing. The man-in-the-middle attack is when cyber criminals intercept communication channels to manipulate transmitted data and falsified operators’ actions. Spooning is when they duplicate data by a third malicious party and send it to the reader after revealing the security protocol. Lastly, phishing is when criminals impersonate trusted and reputable parties to gain critical information such as credit card numbers and passwords.

#### 4.2.2. Opportunities/Solutions

Although the exact shape that smart cities will finally take remains to be indeterminate, scholars agree that there needs to be a variety of precautions, interventions, and solutions to cyber threats to guarantee a smoother implementation process and, ultimately, more secure infrastructure. Some studies included in this SLR provide some solutions and applications that need to be implemented for smart cities to provide some form of security against cyber-criminal activities. For instance, Ref. [41] believe that the current cybersecurity developments for smart cities cannot keep up with the eager adoption of advanced technologies, so there need to be corresponding measures that avert associated cyber threats. In their study, they propose designing correct preventive measures based on deep learning methods to ensure that technological systems in smart cities are robust and well protected. Their paper provides a summary of the knowledge and interpretation of deep learning, cyber security, and smart city concepts, as well as discusses existing related work on IoT security in this setting. Specifically, they review a number of deep learning models that can enhance the security of these cities, including Boltzmann machines, generative adversarial networks, convolutional neural networks, recurrent neural networks, and deep belief networks.

In their paper, Ref. [42] propose investigating the security concern of the smart city’s infrastructure and taking into account the views of both technological and business operations before building a preventive framework. They state that it is vital to analyze the threats before developing safe data. The researchers put forth a Hybrid Smart City Cyber Security Architecture (HSCCA) framework to enhance cybersecurity for smart cities. In this model, they consider key factors such as vulnerable data collection, recovery, memory storage, and well-organized network source supply. To address the key security challenges, their model first highlights and analyzes these problems along with aspects of risks associated with them, and then provides recommendations for solutions and prevention. A similar approach is suggested by ([43,44], and in particular [45]), who design a threat ‘hunting’ model based on Sparse Representation based Classifier (SRC). Their framework identifies cyber threats by Opcode, Bytecode, and system call views to provide suggestions on ways of addressing and preventing such threats.

Reference [46] suggests that, other than focusing on the technical part of the solution, governments should strengthen policies relating to cybersecurity in smart cities. In this paper, the researcher analyzes the impact of China’s 2016 cybersecurity on foreign technology organizations and China’s big data and smart city dreams. He suggests that to reduce threats, cybersecurity and informatization should be seen as ‘two wings, one body’ and must be planned together, arranged together, and moved forward together. Implementation of technologies in smart cities must therefore be integrated with security measures that are built for the long-term. Similar strategies have also been suggested by a number of other scholars, including [47,48] who agree that there should be set standards and avert strategies that govern the implementation of technologies in smart cities to prevent the stated security threats. Such studies also highlight the role of third-party risk management and security ownership in such contexts.

## 5. Discussion

In the context of smart city initiatives in different parts of the world, this SLR reveals that upcoming digital technologies such as AR play a fundamental role in the development of various systems and infrastructures. Indeed, several studies have already been conducted in this area with a keen focus on the application of AR in different areas, including tourism, management, monitoring, education, shopping, transportation, marketing, interior design, and smart parking. Although this exploration remains limited for smart cities, the study identified five main areas where recent scholars have investigated the application of AR for smart cities, including tourism, system monitoring, system management, education and instruction, as well as mobility (aid in movement). Under the tourism category, scholars such as [23,24,25] highlighted some of the areas of AR application for smart cities. Some of these include: smart tourism, smart experience, smart destination, smart trade, geographical information systems, destination managers and marketers, destination image formation, image recognition platforms, multiple viewpoints of the environment, landscape information, scene discovery by image detection, GPS and radar implementation, and prototype screens. Their analyses of these applications are consistent with smart tourism literature in other contexts.

For example, in exploring the value of AR for tourism, Ref. [49] found five value dimensions for AR in tourism, including marketing, organizational, epistemic, touristic, and economical. Ref. [50] also took an internal stakeholder perspective to examine the role of AR in tourism, and found that this technology adds value in this area by modernizing the existing offerings, making it more attractive for new markets. Ref. [51] also suggested the use of AR as robotic tour guides to augment multimedia elements such as 3D objects, sound clips, and video clips to real artifacts in museums. Similarly, Ref. [52] recommended the use of AR games for pre-historical places to enhance tourist interaction. This suggests that AR can be a vital addition for smart cities intending to revamp their tourism sector to attract both domestic and international tourists.

The SLR also discovered that AR could be effectively used for system management and monitoring. Studies included in this research, including [8,26,27,28,29,30], which illustrated how AR could be applied in areas such as real-time information dissemination, smart traffic monitoring, rent management, emergency management, lighting system management, and energy management to eliminate bottlenecks associated in such systems, and in turn, improve efficiency and productivity while reducing costs. Similar studies have also been conducted in other contexts to complement studies included in this SLR. For instance, Ref. [53] found that AR can successfully be used in system management and monitoring in construction, particularly in quality and defects management, time and cost management, safety monitoring and management, worker training, process tracking, and project scheduling. Other researchers that have explored the successful implementation of AR in these areas include [54] (traffic monitoring), Ref. [55] (facility management), and [56] (emergency management), among others.

Education and training are other areas where AR can maximize the potential for smart cities. In this study’s SLR, Refs. [9,31] highlighted how this technology could effectively be used for campus urbanization and assembly-line instruction. Although this area is investigated sparingly in the smart city context, it is one of the most explored in other settings. Researchers such as [57,58,59] have all explored opportunities, challenges, and provided recommendations for the use of AR in education and training. Most of these academic sources agree that the capacity to overlay rich media through AR onto the real world for viewing via web-enabled devices such as smartphones and tablets means that instruction can be made available to learners at the exact time and place of need. This has the benefit of reducing cognitive overload, improving concentration, and making education interesting.

The last category, mobility, is also examined and reported by researchers studying the application of AR in differing frameworks. In the included study for this SLR, Ref. [32] revealed that this digital tech could be used by smart cities to enhance mobility for people with motor disabilities. In line with this, other scholars have also suggested that AR can be used to improve both indoor and outdoor mobility for different people. For example, Ref. [60] recommended an AR indoor positioning system that can be used to track user location and their angles of vision, thereby creating a high adaptability space. Ref. [61] also proposed an indoor navigation system to be used in libraries to help users navigate to shelves and find books. Consistently, Ref. [62] suggested a new AR indoor navigation system for a wheeled robot that can be used in shopping malls and museums. On the other hand, Ref. [63] presented a novel tracking system that can be used for outdoor spaces using beacons. Ref. [64] also proposed a navigation system that can caution drivers about unseen obstacles, and suggested a visual/audio information model powered by AR. For smart cities, such suggestions provide groundwork from which AR-based systems can be implemented to guide navigation and mobility for people.

While technologies such as AR hold great promise for smart cities, the SLR also revealed great concern relating to cybersecurity. Included studies suggest that the interconnectedness of smart cities through advanced technologies gives room for cyber threats. It is suggested that lack of protection against these threats, poor understanding of social engineering, weaponized machine learning technologies by cyber-attackers, non-existent secure device onboarding services, poor encryption key management, as well as lack of cryptographic measures are some of the key issues that contribute to the intensification of cyber threats in smart city ecosystems. Studies included in the SLR, including [34,35,36,37,38], suggest that some of the key challenges in these systems include system and application vulnerabilities, privacy invasion, malware injection attacks, denial of service (DoS), malicious insider threats, and data leakage. Away from smart cities, similar challenges have also been reported by researchers such as [65,66,67,68]. Although solutions to such challenges have been suggested, both technical and non-technical, literature suggests that there is no unanimously agreed solution, and cyber threats will continue to emerge as technologies advance.

## 6. Conclusions and Future Work

This SLR intended to address the study’s main topic: AR and cybersecurity for smart cities. In other words, it intended to investigate the application of AR in this context, and the cybersecurity issues that may arise as a result of the adoption of digital technologies. A review of the final 31 articles was then provided. Based on this review, the application of AR for smart cities can be categorized into five main classifications, including tourism, system monitoring, system management, education and instruction, and mobility (navigation). Tourism, monitoring, and maintenance AR application appear to attract more scholarly attention than education and mobility for smart cities, compared to the general literature on the topic. Regardless of attention, it is believed that a close connection between industry and academic fields is going to be connected. The adoption of these technologies and continued interconnectedness in smart cities is shown to give rise to a host of cybersecurity threats, with among them loss of privacy and confidentiality, physical threats, systems and applications vulnerability, malware injection attacks, denial of service (DoS), malicious insider threats, and data leakage. Solutions to these problems have been suggested by various researchers, but they remain inconsistent and widely unproven. Until now, the majority of studies have attempted to prove concepts rather than describe well-established analytical approaches. In the future, the need for more practical- and analytically based studies is emphasized in order to evaluate discussed hypotheses from this SLR in real smart city contexts.

## Figures and Tables

**Figure 1 sensors-22-02792-f001:**
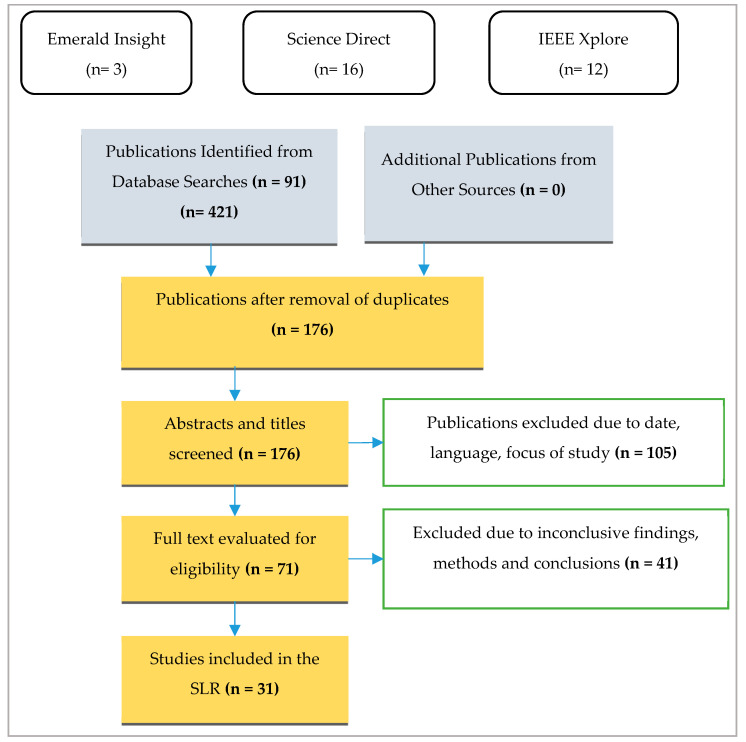
The Preferred Reporting items for SLR flow diagram.

**Table 1 sensors-22-02792-t001:** Key Smart City Components (Alibasic et al., 2016).

Component	Indicators & Benefits
Smart People	Inclusive, creative, educationally excellent
Smart Living	Safety and health, happiness, culturally vibrant
Smart Mobility	Mixed modalities, clean and non-motorized options, ICT, connected
Smart Government	Open data, transparency, e-government application, ICT, supply and demand-side policies
Smart Economy	Local and global business interconnectedness, productivity, innovation, entrepreneurship

**Table 2 sensors-22-02792-t002:** Inclusion and Exclusion Criteria.

Inclusion Criteria	Exclusion Criteria
Addressed the application of AR and cyber-security for smart cites	Addressed related concepts that were not AR and cyber-security in smart cities
Published between 2010 and 2021	Published before 2010
Written in English	Non-English publications
Peer-reviewed journal, conference papers and related publications such as book chapters and seminal works	Unpublished studies (e.g., proposals, theses, and ongoing projects)
Original publications	Duplicates (by title or content)
Publications available online	Publications not available online

## Data Availability

Not applicable.

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
