# Peer review of "Augmented Reality (AR) and Cyber-Security for Smart Cities—A Systematic Literature Review"

_sensors, 2022, doi:10.3390/s22072792_

Round 1
Reviewer 1 Report
In this paper, a systematic literature review fousing on AR and cyber-security for smart city is presented. Overall, the work of this review study is valuable and of significance, and this manuscript is well organized and written. However, the following concerns should be addressed to improve or enhance the work.
- AR is only one theme of digital technologies, comparatively cybersecurity is an more broad theme. Section 4.2 is not closely related to the AR section. Why the paper focuses on the two topics simutaneuously in the context of smart city? May be one of them is fine to researchers and practitioners. Or any other topic should also be reviewed together with the current two topics?
- Generally speaking, 31 key papers seems insufficient as a review paper covering two important topics although the total number of references reach nearly 70 articles.
- How about the literature search using "AR+education"? perhaps there is number of paper can be found. Actually, education has a board scope including daily life, production, studying in schools and universities,etc.
- The authors reviewed the work by HoÅ™ejší, Novikov, and Šimon (2020) as the class of AR asssied instruction. As far as I know, this kind of works in the context of smart factory(within or out of smart city) are numerous. Why the authors only review it? Other key references perhaps should be summarized.
- Please further highlights the findings and suggestions. Some in-depth analysises are encouraged.
- Some works mentioned in the text, e.g. Thomas et al (2017), Yusoff et al.(2018) have not been cited in the references list. Some refeneces are not formal, such as 15 and 27.
Reviewer 2 Report
The article tackles a very interesting topic, combining three different areas of research: AR, cybersecurity and smart cities. In addition, the paper is clear and well-structured. The following points should be taken under consideration to improve the quality of the manuscript:
- The contribution of the article needs to be further explained. Why there is the need to conduct a systematic literature review in this area? The authors briefly describe that there is a lack of research on AR and cybersecurity in the area of smart cities. Although this argument needs to further explained and justified. The authors need to further highlight the contributions of their work in both the abstract and the introduction section.
- An overview of the key findings/outcomes of the review would be helpful within the abstract.
- In terms of completion and in order the work to be more comprehensive and easier to follow, it is advised to provide a small overview of cybersecurity also.
- There are a number of issues pertinent to formatting, i.e., the text, Error! Reference source not found, is found at lines 132, 141.
- The value of table 1 should be provided, in terms of the scope of this work. Also, Table 1 is not cited within the text.
- Methodology: Did the authors follow a standard methodology for conducting their review? It seems yes, but the authors do not mention it. You are strongly advised to provide details on the SLR methodology you followed (standard, initiator, references etc).
- What are the exact keywords the authors used for their queries? It is important to know the keywords used for two reasons: (a) in order to reproduce the SLR process which is the main advantage of the systematic literature reviews against state of the art analysis and (b) in order to evaluate that the keywords and queries are the most suitable ones to address the identified problem.
- “a total of 31 studies met the criteria proposed” (line 207): What are the criteria proposed? The inclusion criteria are used for the initial screening process. What are the criteria used during the full-text read in order to include or not a study?
- Selected databases: There is a plethora of information available and the authors focused on a time period of ten years, which I am sure resulted an enormous number of initial results. Thus, it is understandable that the authors selected only 3 databases (I am guessing to reduce the results). Although, it would be interesting to know the reasons the authors selected the specific databases. For example, why you chose EI over SpringerLink or ACM?
- The authors are advised to revisit some parts of their work. Specifically, the systematic review should follow the PRISMA guidelines as described here: http://prisma-statement.org/
- Identified applications: You discuss several applications of AR in smart cities (Section 4.1), although I think that there are some significant application areas that are not included. For example, healthcare, robotics and the public sector. Why are they missing? From my point of view, I think that this is a disadvantage of the SLR protocol set up. Maybe you need to reconsider the selected keywords and queries.
- The knowledge provided is mainly around the different applications of AR in smart cities, rather than the cyber-security. The contribution related to cyber-security should be enhanced.
- The added value of the Discussion section is somehow limited. The authors lack to critically discuss their findings. They are advised to review their findings and provide their insights related to them.
- Some visualizations (e.g. figures, diagrams, tables) could help understand better the information provided in section 4.2
- Even though section 6 in named “Conclusions and Future Work”, there is not a future work provided. What are you planning to do to further extend this work? What are you planning to do to address some/all of the identified issues? Is there a continuity in this research?
- The citation style should be reviewed and updated according to the instructions. Authors are highly advised to be consistent. Also, it is noted that references must be numbered in order of appearance in the text (including table captions and figure legends) and listed individually at the end of the manuscript. More information can be found here: https://www.mdpi.com/journal/sensors/instructions#references
- The authors are advised to use a more academic language and improve the readability of their manuscript, mainly in terms of grammar and syntax (e.g. line 20-22)
Reviewer 3 Report
The topic of the paper is of great interest and the authors’ review provides useful insights regarding the application of augmented reality and cybersecurity for Smart Cities.
The manuscript has a clear structure, the ideas are organized properly and the methodological approach is well described.
There are only a few errors that should be fixed before publications:
- At lines 190-195 there are several confusions. Please check these lines.
Reviewer 4 Report
The study consists of six parts: Introduction, Smart City Concept, Methods, Results, Discussion, Conclusion & Future Work. The second chapter describes the concept, elements, and general structure of a smart city. The third chapter presents the methodologyadopted in conducting the systematic literature review, including the search strategy and inclusion and exclusion criteria. The fourth chapter presents the results of the review, while the fifth presents the discussion. In the final chapter, the conclusion and future research are stated. This structure is methodically correct, but should be correct item 2 in verse 515 (there was no item 1 in this chapter).
This article briefing provides an overview of AR applications in a smart city.
The authors searched for 421 documents on the basis of keywords in three significant databases of scientific literature (Ei, SD, IX). As a result of deliberate selection, they described the content of 31 studies. Methodology is correct.
The descriptions focus on two areas: AR for Smart Cities (Tourism, System Monitoring, System Management, Education &Instruction, Mobility) and Cyber Security for Smart Cities. The authors pointed the adoption of digital technologies.
Round 2
Reviewer 1 Report
The revisied manuscript has been improved by adressing the comments and concerns from reviewers.
Reviewer 2 Report
Thank you for addressing my comments.
The current version of the review is significantly improved. It is a well organised and presented review in the area of smart cities that has some contribution significance.
The current version of your work I think it is adequate for publishing.